# Key Factors in Crane-Related Occupational Accidents in the Spanish Construction Industry (2012–2021)

**DOI:** 10.3390/ijerph20227080

**Published:** 2023-11-19

**Authors:** Virginia Herrera-Pérez, Francisco Salguero-Caparrós, María del Carmen Pardo-Ferreira, Juan Carlos Rubio-Romero

**Affiliations:** Department of Economics and Business Administration, School of Industrial Engineering, University of Málaga, C/Doctor Ortiz Ramos s/n, 29071 Málaga, Spain; virginia.herrera.p@uma.es (V.H.-P.); carmenpf@uma.es (M.d.C.P.-F.); juro@uma.es (J.C.R.-R.)

**Keywords:** crane, occupational accident, construction of buildings, material agent, safety, accident statistics

## Abstract

The construction industry is one of the riskiest sectors worldwide, with crane operations being one of the most dangerous activities. The aim of this study was to gain insight into the key factors involved in crane-related occupational accidents in the construction industry in Spain. To this end, 1314 accidents involving cranes were analyzed from a total of 241,937 accidents that occurred in the construction of buildings. The data were collected from the Spanish government’s occupational accident statistics corresponding to the years 2012–2021. The results evidenced a statistically significant relationship between cranes as the material agent and the size of the company, with 95% of cases corresponding to small- or medium-sized companies (less than 250 employees). Additionally, it shows how the crane operator is identified as a material contributor to crane accidents in the construction industry, and may be considered a key component to these accidents. In conclusion, improving the knowledge gained about the key factors in crane-related accidents at work in the construction industry provides essential information that helps to design and implement appropriate preventive measures to avoid the recurrence of unwanted events with these machines.

## 1. Introduction

The construction industry is one of the sectors with the greatest impact on the economy of the European Union, and accounts for about 5.7% of its gross domestic product (GDP). Between 2010 and 2021, its contribution to the GDP dropped in 14 Member States, with the largest decreases in Bulgaria, Spain, Greece, and Slovakia, while it grew in other countries such as Hungary, Lithuania, Denmark, Germany, and Finland, which showed the largest rise in that period [1].

On the other hand, occupational accidents in the building industry are very frequent, which makes this sector one of the most dangerous and risky ones worldwide [2,3,4]. The International Labour Organization (ILO) reported more than 1.3 million victims of occupational accidents per year in the construction industry. Moreover, the rate of casualties due to accidents in the workplace in this sector is approximately six cases per 100,000 workers, i.e., three times higher than in any other industry [5]. Therefore, establishing safety measures in every phase of the activity of construction companies should be considered a social priority, with the aim of preventing or reducing occupational risk [6].

In this context, crane operations are among the most dangerous activities in construction because, although crane accidents are not particularly frequent, they are potentially very harmful [7,8]. Thus, 62 workers died in crane-related accidents in Spain between 2012 and 2021 [9], 47 died in Australia between 2003 and 2015 [7], 41 crane-related casualties were recorded in Japan in 2006 [10], and more than 100 tower-crane accidents—resulting in more than 180 deaths and a huge economic loss—were reported in China between 2016 and 2018 [11]. These data are evidence of the safety issues associated with crane operations and how they have become a global challenge [12].

### 1.1. Crane Classification and Types

Moving large or heavy loads in manufacturing or construction requires the use of cranes, which are an essential element for productivity, though subject to several safety concerns [13,14]. Throughout the entire construction phase of a project, cranes are used for most of the vertical or horizontal movements of building resources such as materials, equipment, or personnel [15].

Based on published studies such as that of Shapira et al. [16], cranes can be classified into two machine families: mobile cranes and tower cranes. According to these authors, mobile cranes can be quickly deployed to lift heavy loads, with telescopic booms enhancing versatility and movement capacity. In some regions of North America, the term “mobile cranes” is used for truck-mounted cranes, while track-mounted mobile cranes are considered a separate family, referred to as “crawler cranes”. Tower cranes, on the other hand, are suitable for tight urban construction and reduced spaces, and are available in a variety of sizes and configurations. In recent years, the large growth in real estate development promoted the use of tower cranes in tight construction sites to the point that these cranes have gained worldwide popularity [17].

In Spain, Rubio-Romero and Rubio-Gámez [18] proposed a classification of these machines into the following categories: tower cranes, mobile cranes, light cranes, and overhead cranes; while the Spanish Ministry of Labor and Social Affairs [19,20,21,22] classified them as follows:According to mobilityFixed cranes:
a)supported: attached to a concrete floor plate, rails, walls, etc.;b)embedded: with a base embedded into a concrete foundation.
Mobile cranes:
a)on rails: travelling on rails by its own means of motion;b)climbers: supported on the structure, they grow as the construction progresses.According to the assemblage systemAssembling tower craneSelf-erecting tower crane

### 1.2. Factors Influencing Crane Operation Safety in the Construction Industry

Due to its particular characteristics, the construction industry is affected by inherent factors related to occupational health and safety [23]. The accident rate in the construction industry is among the highest ones worldwide, as compared to other industries or economic sectors [24]. To reduce such rates, it is essential to understand the involved key factors and their influence on health and safety in building projects. In this regard, Mohammadi et al. [2] identified 113 safety-related factors, which they classified into 13 main groups: (1) motivation, (2) rules and regulation, (3) competency, (4) safety investment and costs, (5) financial aspects and productivity, (6) resource and equipment, (7) work pressure, (8) work conditions, (9) culture and climate, (10) attitude and behavior, (11) lessons learned from accidents, (12) organization, and (13) safety programs and management systems. These authors explicitly stated that safety was not only determined by management activities within projects, but also by interactions between factors at different hierarchical levels.

Muñoz-La Rivera et al. [25] proposed a different classification of the factors influencing construction safety. In an interesting PRISMA-based literature review, these authors identified, described, and categorized 100 safety factors grouped in: (1) general aspects, (2) materials and equipment, (3) construction site, (4) human aspects-worker and work team, (5) worker actions/behaviour, and (6) communication. Those 100 identified factors were described and categorized according to the dimensions and aspects of the project that they affected, and were additionally sub-classified as originating factors, shaping factors, or immediate factors, behind the generation of accidents.

As mentioned, analyzing the factors that particularly affect crane-operation safety in the construction industry is essential. In this line, Shapira and Elbaz [26] identified 15 safety parameters related to the crane-operation mode. In their study, favorable and unfavorable work and safety conditions were observed for five working days, half-day from the operator’s cabin and half-day from the ground. From the 15 identified parameters, “Climbing to the cabin” was highlighted, since climbing to the cabin of a tower crane entails great physical effort and a time consumption. This lead some countries, such as France, the Netherlands, Denmark, and Sweden, to establish the regulation that a lift must be used to climb to a cabin higher than 25–30 m.

In a further study focused on crane-operation safety, Milazzo et al. [14] classified the main causes of crane accidents into four groups: (1) electrocutions, (2) blows with the suspended load, (3) collapse of the crane, and (4) blows with the boom. The authors concluded that most accidents were due to improper load handling and poor visibility during load movements.

Zhou et al. [27], in a study based on questionnaires administered to tower crane users, found nine main dimensions and 25 critical factors related to the safety system of tower cranes. This work was based on Rassmussen’s systemic or nonlinear complex conceptual model of accidents (hereinafter Accimap), since tower crane safety concerns are usually systemic or complex issues [28].

In a study by Zhang et al. [11] with an extension in Zhang et al. [29], the authors analyzed 141 crane accident reports from 2013 to 2014 and identified 34 causal factors, which they grouped into six sub-systems: (1) administrative–governmental, (2) tower crane safety management stakeholders, (3) safety management project, (4) staff management, (5) tower crane machinery, and (6) environment management. 

In a further, interesting study based on interviews to experts in construction health and safety, Mohandes et al. [30] identified 21 crane-operation safety factors, which they grouped into four main categories: (1) site conditions, (2) crane operation, (3) site management, and (4) environment. According to these authors, the evidence indicates that crane operation activities are the largest source of accidents in the construction industry, affecting both crane operators and other involved workers.

Finally, Hu et al. [31] proposed an early risk-recognition approach through “Prevention through Design (PtD)”, to manage the exposure to hazards associated with the operation of one or more tower cranes in construction sites, by using a route-finding algorithm and building information modelling (BIM). They provided a taxonomy of crane-related construction risks based on the type of energy involved in the operation and grouped the 15 identified risks into four categories of energy: (1) electricity, (2) motion, (3) gravity, and (4) chemical.

### 1.3. Scope and Contribution of This Research

In the context of the above-mentioned findings, the objective of this study was to gain insight into the key factors involved in cane-related occupational accidents in the construction industry in Spain. To that end, a total of 242,937 cases of construction accidents were analyzed, which included 1314 accidents with the participation of the crane material agent. The data were extracted from the occupational accident statistics of the Spanish Ministry of Labor, Migration, and Social Security, corresponding to the 2012–2021 period. Besides the use of an extensive, updated database, the main novelty in this study was to consider the “crane” as the material agent causing the accidents.

The first step in the proposal of this study is to determine to what extent the degree of injury suffered is related to the associated material agent, in this case the “crane”. Secondly, the aim is to find out to what extent there is a relationship between the material agent “crane” and the age and length of service of the injured person. The third aim is to find out to what extent there is a relationship between the material agent “crane” and whether the company where the injured person worked acted as a contractor or subcontractor. The fourth aim is to find out to what extent the size of the company of the employee involved in the accident is related to the material agent “crane”. This approach makes our study an original contribution, relevant for improving construction health and safety standards and designing strategies aimed at focusing efforts and limiting the serious consequences of this type of accident. 

After this introduction to the context and objectives of the study, the Data and Methods section describes the used sample of occupational accident reports and methodology, including the analyzed variables and statistical analysis, followed by a section of Results and a Discussion of the main findings. Finally, the Conclusions of the study are presented and limitations of the present study.

## 2. Data and Methods

### 2.1. Data Source

In the Member States of the European Union, companies must notify occupational accidents suffered by any of their employees to the corresponding labor authorities in compliance with the regulations of the European Statistics on Accident at Work (ESAW) [32]. Such notification entails codification and registration of the information related to the circumstances of the accident. Since 2003, the Spanish Ministry of Labor and Social Economy registers all accidents that result in one or more days’ sick leave. Such information is harmonized following the guidelines of Directive 89/391/EEC [33] to homogenize the processing of occupational accidents’ data throughout the EU member states. In Spain, the information is recorded on a computer system called Delt@, an acronym for Declaración Electrónica de Trabajadores Accidentados (Electronic Declaration of Injured Workers), which in turn collects the information from occupational accident reports, organized in accordance with Order TAS/2926, of 21 November 2002 [34]. The aforementioned accident reports do not equate to accident investigation reports made by the directly responsible OHS technical advisors in an online investigation.

In Spain, the severity of an occupational accident is evaluated by the physician who manages the victim and reflected in the sick leave certificate [35]. Table 1 shows the total number of occupational accidents in Spain during the 2012–2021 period, according to their degree of severity. A total of 5,506,182 accidents caused sick leave of at least one day.

By using the identification code corresponding to the activity of the involved company, as defined by the CNAE (the national classification of economic activities), which is the Spanish equivalent to the international NACE, CIIU, or ISIC, all accidents occurring in the construction of residential and non-residential buildings (NACE code 41.2) were identified, with a total of 242,937 accidents occurring in Spain in 2012–2021 (Table 2). Crane-related accidents in other economic activities related to the F section “construction” of the NACE Rev.2 classification, such as civil engineering (NACE code 42) or specialized construction (NACE code 43), were out of the scope of this study. The data were collected from the Spanish Government’s Occupational Accident Statistics corresponding to the years 2012–2021.

To analyze occupational accidents related to material agent “crane” in particular, the information in the mentioned database was filtered, classified, and organized, by using the code corresponding to this material agent in the Spanish system for notification of occupational accidents Delt@ [36] and the European Statistics of Accidents at Work (ESAW) of the European Commission [37]. The 1314 crane-related accidents recorded in 2012–2021 were classified according to the severity of injuries and the period required for recovery (Table 3).

### 2.2. Variables

After collecting the information on the total number of accidents in the construction of residential and non-residential buildings in Spain and organizing it according to their degree of severity, we sought to identify factors particularly related to the characterization of those accidents involving cranes.

To analyze crane-related accidents in the workplace regardless of the occupation of the involved worker, this research was designed on certain criteria for screening the data in accident reports. The criteria consisted of selecting all accidents in which a crane was coded as the material agent associated to the specific physical activity, the deviation, or the contact mode of injury. The material agent was further classified using the 8-digit system of CIRCA (Communication and Information Resource Centre for Administrations), where the first four digits of the variable material agent are equivalent to those of the ESAW, and the last four digits provide more detailed information.

Subsequently, variables useful for describing the recorded accidents were selected; thus, our analysis was finally based on the following six variables: (a) codification of the material agent as “crane”, (b) degree of severity of the injury, (c) worker’s length of service, (d) company acting through a contract/subcontract, (e) worker’s age, and (f) company staff (Table 4).

### 2.3. Statistical Analysis

Data were organized and analyzed by using Microsoft EXCEL and the SPSS Statistics V25 software (Statistical Package for the Social Sciences). Besides a descriptive statistical analysis, an inferential statistical analysis was carried out to improve our knowledge on the studied occupational accidents, using the Pearson’s chi-squared test. Variables included in the analysis were pretreated to reduce the originally high number of classes. Thus, classes with higher absolute frequencies were maintained as such, while those with the lower frequencies were grouped under “other”. Subsequently, the significance of possible relationships between different variables was analyzed. In order to select the most suitable test to be used for the inferential analysis, variables were first tested for normality using the Kolmogorov–Smirnov test or the Shapiro–Wilk test, depending on the sample size; results indicated that the distributions were non-parametric. Given that the variables were quantitative, non-parametric, and with multiple categories, the chi-squared test was used to analyze possible associations between different pairs of them. To this end, tests were performed using different pairs of variables that had previously been tested for normality and had shown non-normal distributions. Cases in which a significant association was found between two of these variables were included in the results of the statistical analysis. For this analysis, contingency tables were prepared and the statistical chi-square value (*χ*^2^) was calculated in order to accept or reject the null hypothesis of independence. This statistic associated with a significance level *p* < 0.05 allows us to verify with a confidence level of 95% the relationship of dependence between the variables analysed.

## 3. Results

### 3.1. Severity of the Injury

An analysis of the severity of occupational accidents in Spain in the studied period (2012–2021) showed that the number of accidents grew throughout the years in all sectors, in the construction of buildings and in crane-related events, with the highest figures in the year before the COVID-19 pandemic, which was 2019.

Figure 1 shows the relative frequencies of the different degrees of severity of accidents in Spain in the studied period. From the total occupational accidents, 98.91% were minor, 0.82% were serious, and 0.12% were fatal. The frequencies were similar in the construction of buildings with 98.51% minor accidents, 1.27% serious accidents, and 0.13% casualties. However, the relative frequencies of notified accidents involving cranes were slightly different, with 91.32% minor accidents, 2.05% serious accidents, and 0.61% casualties. This latter finding could be influenced by the fact that 5.94% of the corresponding accident reports did not provide information on the severity.

### 3.2. Material Agent

The harmonized variable material agent of the ESAW coding system for the analysis of accidents includes the following sub-categories: material agent of the specific physical activity, material agent associated with the deviation, and material agent associated with the contact mode of injury [37]. The first one is defined as the tool, object or instrument used by the victim immediately before the accident, the second one is the tool, object, or instrument involved in the abnormal event (the deviation), and the third one is the object, instrument, or tool with which the victim came into contact.

Accidents involving cranes as the material agent associated to the specific physical activity, the deviation or the contact mode of injury amounted to 1314 cases. Figure 2 shows that the evolution of the number of accidents was uneven throughout the studied period, January 2012 to December 2021 (Figure 2).

The eight-digit codification of crane-related material agents yielded 10 different subtypes (Table 5). Remarkably, code 11.03.01.01 “cranes” accounted for one-third of notified accidents in all the three material agent associations. It was also noticeable that almost a quarter of all accidents included code 14.11.00.00 “load suspended from a hoisting device, a crane”. Most remarkable was code 11.03.01.02 “load/unload handler” with an average of 13.27 of notified cases, which indicated that the crane operator was identified as a material agent contributing to this type of accident.

### 3.3. Length of Service

Workers’ length of service was analyzed using the following categories: up to 1 month, 1 to 2 months, 3 to 4 months, 5 to 7 months, 8 to 12 months, 1 to 3 years, 3 to 10 years, and more than 10 years. As shown in Figure 3, most injured workers had up to 1 month experience, with a relative frequency of 20.93%. Moreover, 63.33% of notified crane accidents involved workers with less than 1 year experience, as compared with 8.22% of accidents corresponding to workers with more than 10 years.

### 3.4. Company Acting through a Contract/Subcontract

The contractor/subcontractor role of involved companies was analyzed from accident reports. In these reports, “Yes” had to be marked if, at the moment of the accident, the worker was paying service as part of a contractor/subcontractor company, namely their company was in charge of tasks related to the own-activity of a third company. The general concept “own-activity” does not include hiring companies dedicated to the cleaning, maintenance, security, or repair/extension of premises. The results showed than only one-third of reports stated that the victim was working in a company that acted as a contractor/subcontractor in the construction of buildings (Figure 4).

### 3.5. Age of the Injured Worker

Accidents were then analyzed according to the age of injured workers, classified into the following ranges: up to 24 years of age, from 25 to 34 years, from 35 to 44 years, from 45 to 54 years, and 55 or more years. Figure 5 shows that most of the injured workers were between 45 and 54 years old, followed by workers from 35 to 44 years old, with frequencies of 33.87% and 32.19%, respectively.

### 3.6. Company Staff

This analysis was focused on the size of the company injured workers were working for, expressed as the number of workers, and classified into the following ranges: less than 5 workers, 6 to 10 workers, 11 to 25 workers, 26 to 50 workers, 51 to 100 workers, 101 to 250 workers, and more than 250 workers. Figure 6 shows that 95% of crane accidents occurred in companies with less than 250 workers. Remarkably, 52.59% of them occurred in companies of less than 25 workers.

### 3.7. Inferential Statistical Analysis

In the inferential statistical analysis, possible associations between different variables were assessed. Table 6 summarizes the main results.

The Pearson’s chi-squared test indicated no significant association between the severity of the injury and the material agent, associated either to the specific physical activity, the deviation or the contact mode of injury; although the relationship between the severity and the material agent associated to the specific physical activity showed the most stable value (*p* = 0.118).

Similarly, the analysis of the relationship between the material agent and the worker’s length of service failed to show any significant association, with the material agent associated to the contact mode of injury being the closest one to significance (*p* = 0.171). 

The analysis of the variable company-acting-through-a-contract/subcontract showed a significant association with the material agent associated to the specific physical activity (*p* < 0.001), while no significance was found for the relationship with the material agent associated to the deviation (*p* = 0.289) or the contact mode of injury (*p* = 0.170), for which the null hypothesis was accepted. 

Regarding the injured worker’s age, the Pearson’s chi-squared test failed to reveal any significant association with the material agent; thus, the null hypothesis was accepted. Finally, the size of the company was significantly associated to all the material agent associations, i.e., with the specific physical activity, the deviation and the contact mode of injury (*p* < 0.001); thus, the null hypothesis was rejected.

## 4. Discussion

Identifying and coding the variable material agent in any of its associations is critical to establishing the circumstances under which an accident occurred [42,43,44,45,46,47,48]. However, few published studies have used this variable in the analysis of occupational accidents in the construction industry. Among them, Camino-López et al. [39] analyzed a total of 1,630,452 accidents suffered by construction workers in Spain between 1990 and 2000; Betsis et al. [49] analyzed a sample of 413 notified construction accidents that occurred in the north of Greece between 2003 and 2007; and finally, Fontaneda et al. (2022) [50] analyzed 455,491 construction accidents in Spain notified between 2011 and 2018. Only the mentioned study by Camino-López et al. [39] included “cranes and lifting equipment” among the analyzed material agents.

From our results and analysis of the 10 subtypes of a material agent crane, it is remarkable that “loading/unloading manipulator”, which might be assimilated to the crane operator, is a factor contributing to this type of accident in 13.27% of notified cases. The relevance of the crane operator as a key factor in safety operations with these machines was evidenced in studies such as Zhou et al. [27], who considered that it was the third most important factor out of 25 that directly affect the safety of tower crane operations. Mohandes et al. [30] also identified the “crane operator skills” as a safety factor in crane operations in construction projects. Such skills are directly linked to the training that a crane operator has received; hence, authors such as Manzoor et al. [51] identified “non-qualification of the crane operator” as a contributing factor in accidents occurred in the construction of high-rise buildings.

Also noteworthy is the relationship between crane accidents and the company acting through a contract/subcontract. In recent years, outsourcing in the construction industry has experienced a dramatic unprecedented growth. This phenomenon had negative effects resulting in violations of the occupational health and safety regulations, which contributed to higher rates of accidents in the workplace and promoted poor working conditions, often due to a cascade of subcontractors [52,53]. Some authors, e.g., Choudhry et al. [54], proposed that outsourcing in the construction industry should be regulated with the aim of reducing the number of subcontractor levels to effectively manage the communication gap between the main contractor and the subcontractor. Furthermore, subcontracting has adverse influences on the health and safety of construction workers [55]. Therefore, in addition to reducing the number of subcontractor levels, working with a regular chain of subcontractors, accredited and controlled by the corresponding occupational safety authority, should be encouraged [56]. In this line, Zhang et al. [11] reported that one of the factors particularly affecting crane operation safety in the construction industry was misuse of safety regulations, such as subcontracting out of regulations, as described by Tam and Fung (2011) [57]. The latter authors investigated tower crane safety in relation to the understanding and degree of observation of legal requirements, as well as non-legal practices in the Hong Kong construction industry. They found that indolent performance of tower crane operation professionals was one of the main causes behind unsafe practices related to these tasks.

In Spain, Law 32/2006 of 18 October, which regulates subcontracting in the construction industry [58], addressed this situation by, among other things, restricting the number of subcontractor levels in this area. This Law regulates the subcontracting regime and seeks to eliminate subcontracts that are economically unproductive or harmful for workers’ health and safety. As a complement to the Subcontracting Law 32/2006, Royal Decree 1109/2007 of 24 August, aimed to establish the rules for the application and development of the aforementioned regulatory law in the construction sector [59]. Thus, it was particularly perplexing that up to 69% of the crane accident reports analyzed in this study indicated that the workers were not paying services with a contractor/subcontractor, given that all construction work must be integrated into one of the legally established contractor/subcontractor levels. This situation could be accounted for by errors in completing the accident notification for the labor authorities, something that may happen if the person who completes the notification, whether administrative or technical staff, does not work for the same company as the injured worker or has poor training or knowledge of the correct way to complete these reports. Authors such as Salguero-Caparrós et al. [46], Molinero-Ruiz et al. [47], and Jacinto et al. [60] also found that errors completing and coding accident reports were common.

Regarding workers’ length of service, more than two-thirds of the notified accidents occurred to workers with less than one year experience. This finding is in line with the results of Camino-López et al. [39] and Betsis et al. [49], who reported that 68.1% and 68% of notified accidents, respectively, involved workers with less than one year experience. This aspect may be linked to the high temporality, workplace turnover, and successive job changes that are rather frequent in this sector and is one of the best-known risk factors for crane accidents [11]. 

Regarding the worker’s age, up to two-thirds of accidents involved 35–54-year-old subjects. This result cannot be compared with other studies, since no studies analyzing the age of the injured worker are available. We would like to highlight the positive finding that only 3.58% of accidents involved workers younger than 24 years. Length of service and age are important factors to be considered in the interpretation of crane-related accidents [7], and full comprehension of the roles of both personal variables in this type of analysis requires further research. In this way, it may be possible to reduce the gaps in our knowledge of this subject and their serious consequences when it comes to improving health and safety.

An analysis of the size of the companies involved in crane-related accidents revealed that smaller companies accumulated more accidents. Based on the definition provided by the European Commission [61], companies with less than 250 employees were considered small- and medium-sized companies, including micro-companies (4–9 employees), small companies (10–49), and medium-sized companies (50–249). The results of this study (2012–2021) showed that crane operators working for micro-companies suffered 27.7% of all crane-related accidents, those working for small companies suffered 43.92% and those belonging to medium-sized companies accounted for 23.44% of the total reported cases. These data were in agreement with those of Camino-López et al. [39] and Fontaneda et al. [50], both related to the Spanish construction industry. The analysis of the relationship between the size of the company and the rate of crane-related occupational accidents evidences an important, largely neglected risk factor of psychological nature; namely, operating a crane is more sophisticated and mentally demanding than operating other equipment; therefore, crane operators may be more vulnerable to human error [62].

## 5. Conclusions

Our results suggest that both occupational accidents in the construction of buildings and occupational accidents related to the material agent crane have progressively increased in the last decade in Spain. In particular, a statistically significant relationship was observed between the material agent crane and the small or medium size of the construction company.

Small- and medium-sized companies are one of the strengths of the European economy, with up to 60% of the total workforce and key resources for innovation, growth, and productivity. In addition, the construction industry directly impacts the national economy and plays an important role in its growth. On the other hand, the construction industry is associated to high accident rates, as evidenced by the alarming statistics. Furthermore, since cranes are one of the most frequently used equipment in construction sites, and are also associated to high accident rates, preventing crane accidents is an urgent mission. This situation was the main reason behind the present research, focused on the key factors related to crane accidents in the construction industry. 

This study also evidenced that 63.33% of workers suffering accidents with cranes in Spain in the studied period had no more than one year experience, which suggests that longer time working in the company entails more training and experience and may reduce the probability of suffering an accident. However, it should be kept in mind that experienced workers, who have become highly familiar with the involved tasks, may tend to disregard the danger and underestimate the risks. 

This study also showed how the crane operator is identified as a material contributor to crane accidents in the construction industry, and may be considered a key component to these accidents. Aspects such as long working hours, insufficient rest breaks, working under pressure due to tight deadlines, communication failures between the crane operator and the signalman of the operations or inadequate training of the crane operator support the proposal that human factors are relevant to crane-operation safety. Naturally, technical issues are one of the most dangerous risk factors with cranes and should not be disregarded, although the human factor is a priority that requires the implementation of corrective actions.

In conclusion, the different approaches to safety-based decision-making associated with crane operations show several shortcomings, including the lack of an inclusive approach to identifying critical causal factors. Besides using a holistic inclusive approach, such factors should also be addressed in a specific way, since cranes are complex installations that constitute a critical aspect of safety in construction sites. In summary, improving our knowledge of the key factors in crane-related occupational accidents in the construction industry provides essential information to design and implement suitable measures to prevent unwanted events with these machines. Finally, we propose that efforts should be devoted not only to negative aspects such as accidents, human error, or technical issues, but also to positive aspects of the widely variable daily performance of a system as complex as that of the construction industry.

### Limitations

The abovementioned conclusions should be seen in the light of the limitations of the present study. First, possible errors in coding and fulfilling the accident reports; second, the lack of specific data concerning the description of the occupational accidents, although the results of our study are of interest for the whole of the Spanish construction industry; third, the data source might be incomplete because, although all accidents in the construction of residential and non-residential buildings notified in 2012–2021 were analyzed, there might be more accidents that were not notified; finally, accidents that did not result in the loss of working days are not notified as occupational accidents.

## Figures and Tables

**Figure 1 ijerph-20-07080-f001:**
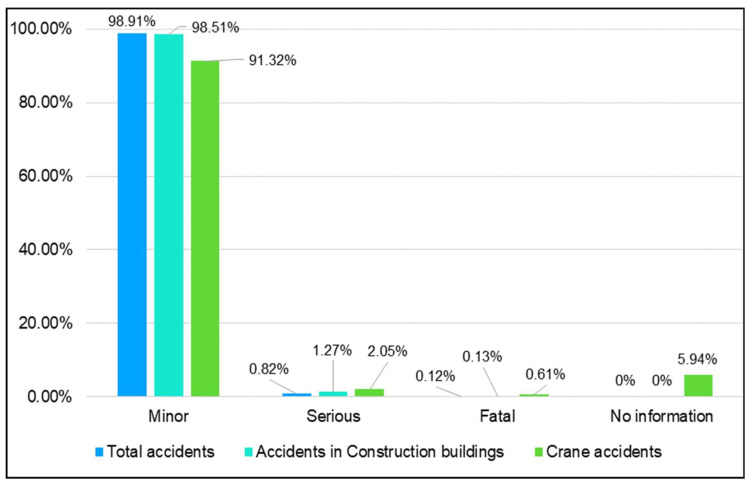
Distribution of occupational accidents with sick leave according to severity in Spain in 2012–2021. Classification according to total accidents, accidents in building construction, and accidents with cranes.

**Figure 2 ijerph-20-07080-f002:**
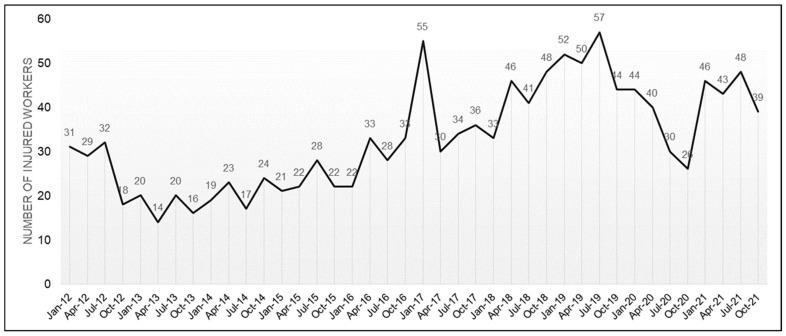
Distribution of accidents involving cranes in Spain between January 2012 and December 2021.

**Figure 3 ijerph-20-07080-f003:**
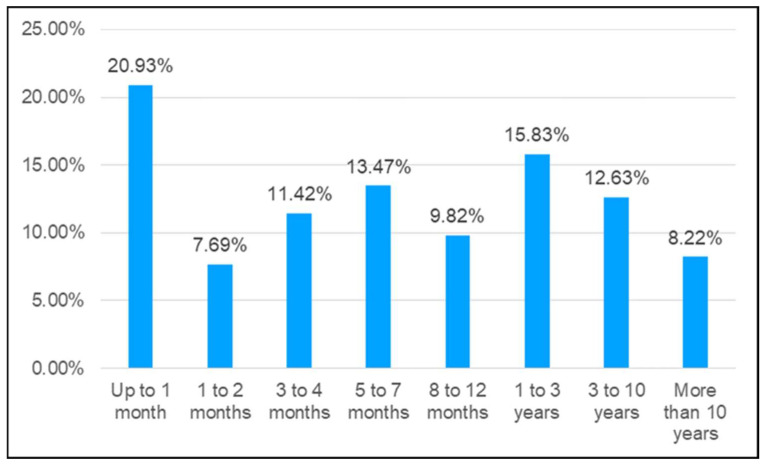
Frequency of accidents involving cranes in Spain 2012–2021 according to length of service in the workplace.

**Figure 4 ijerph-20-07080-f004:**
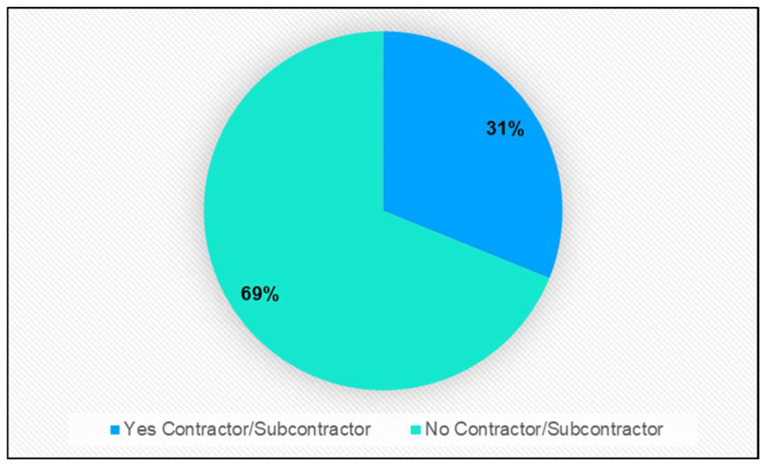
Frequency of accidents involving cranes in Spain 2012–2021 according on where they provided their services.

**Figure 5 ijerph-20-07080-f005:**
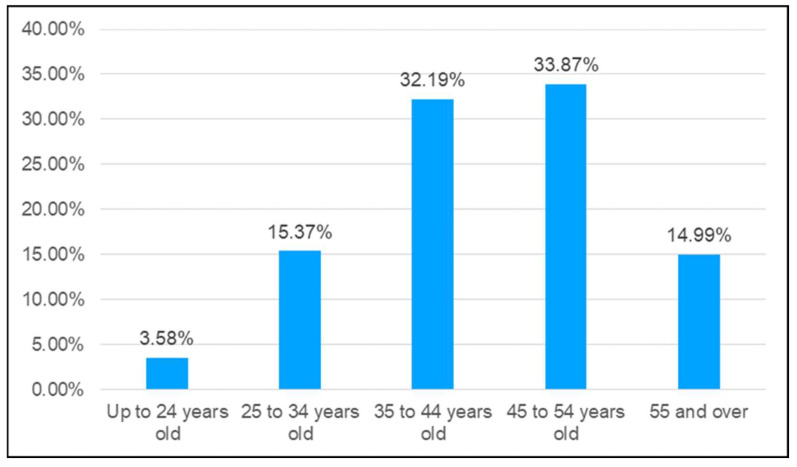
Frequency of accidents involving cranes in Spain 2012–2021 according to the injured worker’s age.

**Figure 6 ijerph-20-07080-f006:**
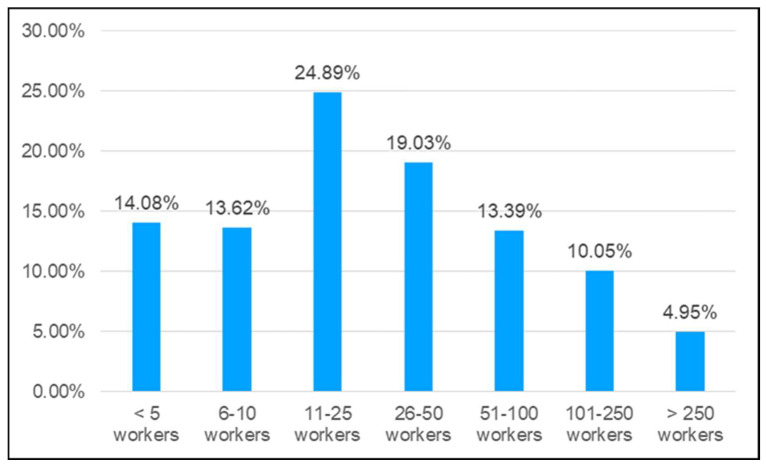
Frequency of accidents involving cranes in Spain 2012–2021 according to the company staff.

**Table 1 ijerph-20-07080-t001:** Total occupational accidents with sick leave in Spain during the period 2012–2021 classified per severity.

Year	Total	Minor	Serious	Fatal
2012	471,223	466,953	3798	472
2013	468,030	464,153	3420	457
2014	491,099	486,306	4213	580
2015	529,248	524,210	4409	629
2016	575,235	560,957	4649	629
2017	596,606	591,009	4968	629
2018	617,488	611,727	5032	729
2019	650,602	644,709	5394	721
2020	505,528	500,299	4474	755
2021	601,123	595,616	4796	742
TOTAL	5,506,182	5,445,939	45,153	6343

**Table 2 ijerph-20-07080-t002:** Occupational accidents in building construction with sick leave in Spain during the period 2012–2021 according to severity.

Year	Total	Minor	Serious	Fatal
2012	20,357	20,015	316	26
2013	16,181	15,939	22	4
2014	16,351	16,101	225	25
2015	18,771	18,473	272	26
2016	21,144	20,819	293	32
2017	24,970	24,611	329	30
2018	29,240	28,828	372	40
2019	35,151	34,626	466	59
2020	28,464	27,995	426	44
2021	32,308	31,899	369	40
TOTAL	242,937	239,306	3090	326

**Table 3 ijerph-20-07080-t003:** Total occupational accidents related to cranes with sick leave in Spain during the period 2012–2021 according to severity.

Year	Total	Minor	Serious	Fatal	No Information
2012	109	94	3	1	11
2013	70	59	2	1	7
2014	83	72	4	0	8
2015	93	84	3	0	6
2016	116	104	3	0	9
2017	155	143	4	2	6
2018	167	152	2	0	13
2019	205	188	4	2	11
2020	139	135	0	1	3
2021	177	169	2	1	5
TOTAL	1314	1200	27	8	79

**Table 4 ijerph-20-07080-t004:** Variables used in the analysis of crane accidents.

Variable	Categories	Reference
Material Agent	(11.03.00.00) Fixed, mobile, vehicle-mounted, overhead, overhead cranes, levelling equipment	ESAW Variable [37]
	(11.03.01.00) Cranes, bridge cranes	ESAW Variable [37]
	(11.03.01.019 Cranes	ESAW Variable [37]
	(11.03.01.02) Loading/unloading manipulator	ESAW Variable [37]
	(11.03.01.03) Overhead and gantry cranes	ESAW Variable [37]
	(11.03.01.04) Vehicle loading arm	ESAW Variable [37]
	(11.03.02.00) Winches, hoists, balancers	ESAW Variable [37]
	(11.03.02.01) Winches, hoists, lifting pulleys, mufflers, balancers	ESAW Variable [37]
	(11.03.99.00) Other overhead lifting equipment	ESAW Variable [37]
	(14.11.00.00) Loads—suspended from leveling device, a crane	ESAW Variable [37]
Severity of injury	(1) Minor	Fuentes-Bargues et al. [38]
	(2) Serious	Fuentes-Bargues et al. [38]
	(3) Fatal	Fuentes-Bargues et al. [38]
Length of service	(1) Up to 1 month	Camino-López et al. [39]
	(2) 1–2 months	Camino-López et al. [39]
	(3) 3–4 months	Camino-López et al. [39]
	(4) 5–7 months	Camino-López et al. [39]
	(5) 8–12 months	Camino-López et al. [39]
	(6) 1–3 years	Camino-López et al. [39]
	(7) 3–10 years	Camino-López et al. [39]
	(8) More than 10 years	Camino-López et al. [39]
Contracts or subcontracts	(1) Yes contractor/subcontractor	Salguero-Caparrós et al. [40]
	(2) No contractor/subcontractor	Salguero-Caparrós et al. [40]
Age of the injured worker	(1) Up to 24 years old	Fontaneda et al. [41]
	(2) 25–34 years old	Fontaneda et al. [41]
	(3) 35–44 years old	Fontaneda et al. [41]
	(4) 45–54 years old	Fontaneda et al. [41]
	(5) 55 years or older	Fontaneda et al. [41]
Company staff	(1) <5 employees	Fuentes-Bargues et al. [38]
	(2) 6–10 employees	Fuentes-Bargues et al. [38]
	(3) 11–25 employees	Fuentes-Bargues et al. [38]
	(4) 26–50 employees	Fuentes-Bargues et al. [38]
	(5) 51–100 employees	Fuentes-Bargues et al. [38]
	(6) 101–250 employees	Fuentes-Bargues et al. [38]
	(7) >250 employees	Fuentes-Bargues et al. [38]

**Table 5 ijerph-20-07080-t005:** Frequency distribution of accidents per material agent associated with the specific physical activity, the deviation, or the contact mode of injury.

		Associated to the Specific Physical Activity		Associated to the Deviation		Associated to the Contact Mode of Injury	
Code	Material Agent	Absolute Frequency	Relative Frequency (%)	Absolute Frequency	Relative Frequency (%)	Absolute Frequency	Relative Frequency (%)
11.03.00.00	Fixed, mobile, vehicle-mounted, overhead, overhead cranes, levelling equipment	180	17.27%	151	16.01%	141	16.02%
11.03.01.00	Cranes, bridge cranes	36	3.45%	31	3.29%	29	3.30%
11.03.01.01	Cranes	327	31.38%	273	28.95%	245	27.84%
11.03.01.02	Loading/unloading manipulator	141	13.53%	130	13.79%	110	12.50%
11.03.01.03	Overhead and gantry cranes	3	0.29%	4	0.42%	5	0.57%
11.03.01.04	Vehicle loading arm	22	2.11%	29	3.08%	18	2.05%
11.03.02.00	Winches, hoists, balancers	4	0.38%	1	0.11%	1	0.11%
11.03.02.01	Winches, hoists, lifting pulleys, mufflers, balancers	27	2.59%	25	2.65%	22	2.50%
11.03.99.00	Other overhead lifting equipment	33	3.17%	25	2.65%	30	3.41%
14.11.00.00	Loads—suspended from levelling device, a crane	269	25.82%	274	29.06%	279	31.70%
	Total	1.042	100%	943	100%	880	100%

**Table 6 ijerph-20-07080-t006:** Main results of the inferential statistical analysis.

Independent Variable	Dependent Variable	Pearson’s Chi-Square. Significance Level (*p*-Value)	Pearson’s Chi-Square (*χ*^2^)
Severity of injury	Material agent associated with specific physical activity	0.118	25.264
	Material agent associated with the deviation	0.365	19.451
	Material agent associated with the contact mode of injury	0.220	22.272
Length of service	Material agent associated with specific physical activity	0.227	71.052
	Material agent associated with the deviation	0.794	53.612
	Material agent associated with the contact mode of injury	0.171	73.533
Contracts or subcontracts	Material agent associated with specific physical activity	<0.001	16.674
	Material agent associated with the deviation	0.289	10.806
	Material agent associated with the contact mode of injury	0.170	12.839
Age of the injured worker	Material agent associated with specific physical activity	0.739	30.237
	Material agent associated with the deviation	0.353	38.585
	Material agent associated with the contact mode of injury	0.268	40.783
Company staff	Material agent associated with specific physical activity	<0.001	127.638
	Material agent associated with the deviation	<0.001	108.629
	Material agent associated with the contact mode of injury	<0.001	108.642

## Data Availability

Data available on request at the Spanish Government’s Occupational Accident Statistics.

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
