# Peer review of "Key Factors in Crane-Related Occupational Accidents in the Spanish Construction Industry (2012–2021)"

_ijerph, 2023, doi:10.3390/ijerph20227080_

Round 1

Reviewer 1 Report

Comments and Suggestions for Authors

The paper provides valuable insights into the key factors involved in crane-related occupational accidents in the Spanish construction industry, based on the analysis of a significant number of accidents over a nine-year period. From my point of view, the paper should be accepted after some revisions as follows: 

The paper could be improved by providing a more detailed analysis of specific crane types, causes of accidents, and exploring additional influencing factors;

Provide a more detailed analysis of the specific types of cranes involved in the accidents, as this could offer insights into the factors contributing to accidents associated with different crane types;

- Further details on the data collection, selection criteria, and variable definitions would enhance the robustness of the methodological description; 

- The summarized sections do not provide insights into the detailed statistical analysis methods used or specific recommendation based on the findings;

Overall, while the paper provides valuable insights, it could benefit from further exploration of specific crane types, causes of accidents, additional influencing factors, as weather conditions, maintenance practices, etc.

Author Response

Ref.:  IJERPH-2652966

Titled: "Key factors in crane-related occupational accidents in the Spanish construction industry (2012-2021)”

Response to Editor and Reviewers

Dear Reviewer   

I would like to thank you for your accurate and detailed revision for improving the paper. In the attached file, you will find a new version of the article, which has been revised in accordance with the recommendations of the editor and reviewers. New comments and explanations according to editor’ suggestions and reviewers’s suggestions have been highlighted in blue (Reviewer #1), yellow (Reviewer #2), or green (Reviewer #3) based on who suggested the changes.  In what follows, we explain how we have responded to and dealt with each of the comments.

Reply to Reviewer #1:

Reviewer #1: The paper provides valuable insights into the key factors involved in crane-related occupational accidents in the Spanish construction industry, based on the analysis of a significant number of accidents over a nine-year period. From my point of view, the paper should be accepted after some revisions as follows

- The paper could be improved by providing a more detailed analysis of specific crane types, causes of accidents, and exploring additional influencing factors;

- Provide a more detailed analysis of the specific types of cranes involved in the accidents, as this could offer insights into the factors contributing to accidents associated with different crane types;

- Further details on the data collection, selection criteria, and variable definitions would enhance the robustness of the methodological description; 

- The summarized sections do not provide insights into the detailed statistical analysis methods used or specific recommendation based on the findings;

Overall, while the paper provides valuable insights, it could benefit from further exploration of specific crane types, causes of accidents, additional influencing factors, as weather conditions, maintenance practices, etc

Thank you very much for your very constructive comments which will surely help us to improve the article.

In Spain, when there is an accident at work that results in one or more days absence from work, the company where the injured worker worked must generate a notification or report of the accident at work through the computer system called "Delt@".

The aforementioned accident report is made up of 58 variables in 7 different sections.

These 7 sections are:

1.- Data on the injured worker

2.- Details of the work centre where the injured worker is affiliated.

3.- Data on the place and centre where the accident took place.

4.- Details of the circumstances of the accident

5.- Details of the care provided

6-  Data on the end of the Temporary Incapacity process.

7.- Data on the work centre where the injured worker is affiliated.

As you can see, this is a standardised electronic declaration of the accident data, but in no way can it be considered as an in-depth accident investigation report. However, these 58 variables provide extremely valuable information for the subsequent investigation of the accident at work.

This is why data such as the causes of the accident, factors influencing the accident such as environmental conditions, weather conditions and maintenance practices have not been included in this study.

In order to briefly clarify this aspect, the following paragraph is modified in the Section 2.1:

“In Spain, the information is recorded on a computer system called Delt@, an acronym for Declaración Electrónica de Trabajadores Accidentados (Electronic Declaration of Injured Workers), which in turns collects the information from occupational accident reports, organized in accordance with Order TAS/2926, of November 21st, 2002 [36]. The aforementioned accident reports do not equate to accident investigation reports made by the directly responsible OHS technical advisors in an online investigation.”

On the other hand, with regard to the specific types of cranes, in this study we proceeded to select, classify and tabulate the variables corresponding to the material agent, whether it was associated with the specific physical activity, the deviation or the form of contact always related to the material agent crane. The 8-digit coding used by the CIRCA system in Spain was applied as opposed to the 4-digit coding used by ESAW in Europe, which meant that we went from two types of material agent associated with cranes to up to 10 different types. But even so, this classification does not show sufficient detail as to the specific type of crane involved in the accident that occurred.

Reviewer 2 Report

Comments and Suggestions for Authors

This could be a publishable paper, but some weaknesses identified in the attached report prevent me from recommending it at this stage.

Comments on the Quality of English Language

There were really very few mistakes in this paper that I could detect; they are identified in the attached report.

Author Response

Ref.:  IJERPH-2652966

Titled: "Key factors in crane-related occupational accidents in the Spanish construction industry (2012-2021)”

Response to Editor and Reviewers

Dear Reviewer   

I would like to thank you for your accurate and detailed revision for improving the paper. In the attached file, you will find a new version of the article, which has been revised in accordance with the recommendations of the editor and reviewers. New comments and explanations according to editor’ suggestions and reviewers’s suggestions have been highlighted in blue (Reviewer #1), yellow (Reviewer #2), or green (Reviewer #3) based on who suggested the changes.  In what follows, we explain how we have responded to and dealt with each of the comments.

Reply to Reviewer #2:

Reviewer #2: I quite agree with the authors when they write “that safety issues associated with crane operations have become a global challenge”. That is why this paper is very relevant. However, I strongly disagree with the authors when they write that “This study also evidenced the influence of the human factor, in particular that related to the crane operator, as a key component of accidents in the construction industry”. Besides “age” and “length of service”, no significant human factor variable was processed in this study.

Thank you very much for your very constructive comments which will surely help us to improve the article. We will try to respond to all your comments below:

In relation to your assessment, the sentence is modified as follows in the Abstract of the paper:

“Additionally, it shows how the crane operator is identified as a material contributor to crane accidents in the construction industry, and may be considered a key component to these accidents”.

In general, this is a good paper, that has quite a few good features to it:

  • The review of the literature is one of the finest I have seen in quite some time;
  • The methodology is quite adequate, and very clearly described;
  • With the few exceptions mentioned below, it is very well written; the quality of language is good;
  • Also, the authors are well aware of the limitations of their research.

The weaknesses of this paper are as follows:

  • In page 2, there is a discussion about the various types of cranes; one would expect this to be used as one of the variables in the analysis; from a prevention point of view, the hypothesis that some types of cranes are more dangerous than others would make sense; yet the authors provide no data or results about this; were the data about the specific type of crane unavailable in the database?

It was considered very necessary to provide a selection and classification of the different types of existing cranes according to experts in the field. However, the notification or report of a work accident through the computer system called “Delt@” that is carried out in Spain does not offer specific information on the type of crane that was being used in the reported accident, beyond the material agent included in this studio.

  • There is an indication of this in Table 5; the two most frequent material agent types are “Fixed, mobile, vehicle mounted etc.” (Code 11.03.00.00) and “Crane (Code 11.03.01.01); this aspect of the results is underexploited by the authors;

As you can see, this is a standardised electronic declaration of the accident data, but in no way can it be considered as an in-depth accident investigation report. However, these variables provide extremely valuable information for the subsequent investigation of the accident at work in an appropriate manner. Even so, as far as the specific types of cranes are concerned, in the present study, the variables corresponding to the material agent were selected, classified and tabulated, whether associated with the specific physical activity, the deviation or the contact-mode of injury, always related to the material agent crane. The 8-digit coding used by the CIRCA system in Spain was applied as opposed to the 4-digit coding used by ESAW in Europe, with which we went from two types of material agent associated with cranes to up to 10 different types.

  • Yet, in the discussion section, they come up with the statement that: “From our results and analysis of the 10 subtypes of material agent crane, it is remarkable that “loading/unloading manipulator”, which might be assimilated to the crane operator, is a factor contributing to this type of accidents in 13.27% of notified cases”; this is a dangerous assumption which overlooks 1) two more important categories mentioned above, and 2) there could be other underlying factors such as a possible communication problem between the operator and the signal person, for example; that is why I disagree that their study evidenced the importance of the human factor;

In line with your suggestion, the following paragraph in the Conclusions section is amended as follows:

“This study also showed how the crane operator is identified as a material contributor to crane accidents in the construction industry, and may be considered a key component to these accidents.Aspects such as long working hours, insufficient rest breaks, working under pressure due to tight deadlines, communication failures between the crane operator and the signalman of the operations or inadequate training of the crane operator support the proposal that human factors are relevant to crane-operation safety”.

  • Is there a reason why cranes used in civil engineering were excluded? If so, it should be explained;

As explained in the methodology section, accidents with cranes in civil engineering and specialised construction activities were left out of the scope of this study, since the aim was to eliminate accidents in transit and in itinere that can sometimes occur in civil engineering.

  • In line 164 the authors mention that in the conclusion “future research lines are proposed”; however, in effect the research propositions are somewhat fuzzy;

Following your suggestion and in order to clarify this aspect, the final paragraph in the introductory section is amended:

“Finally, the Conclusions of the study are presented and limitations of the present study”

  • I am not surprised that the variables “age of the injured worker” and “length of service” did not prove significant; generally, these variables don’t provide useful information for prevention purposes; it must be said, however, that the fact that a large majority of accidents occurred to employees with less than 1 year experience might suggest a need for major training improvement;

I very much agree with your assessment. This is why the third paragraph of the conclusions states that:

This study also evidenced that 63.33% of workers suffering accidents with cranes in Spain in the studied period had no more than one year experience, which suggests that longer time working in the company entails more training and experience and may reduce the probability of suffering an accident. However, it should be bore in mind that experienced workers, who have become highly familiar with the involved tasks, may tend to disregard the danger and underestimate the risks.

  • The last paragraph of the conclusion seems to imply that the database was rather poor as regards critical contributing factors; is that the case? In fact a better description of the contents of the database, earlier in the paper, would be useful;

The data used in this study come from the Spanish Government's Occupational Accident Statistics for the years 2012 to 2021. This is a very important and essential database for both the Spanish administration and society. This is why we do not consider it to be deficient, although it can be improved, like any database.  This is the aspect that we wanted to highlight in the limitations section.

  • One third of the workers were working for contractors/subcontractors; who were the other two thirds working for? The site/project owner? It would be worth clarifying this point in lines 335-337

Thank you for your appreciated suggestion. This is one of the aspects that most caught our attention. That is why the article has expanded this debate with the following paragraph:

“This situation could be accounted for by errors in completing the accident notification for the labor authorities, something that may happen if the person who completes the notification, whether administrative or technical staff, does not work for the same company as the injured worker or has poor training or knowledge of the correct way to complete these reports.”

There were few minor mistakes that I could identify:

  • Line 142: “crane-related” (letter missing)
  • Page 7: the separation lines between variables are missing or misplaced
  • Page 7: last line in the “Length of service” category should read “More than…”
  • Line 263: is it a coincidence that the percentage of fatal accidents is equal to the percentage of reports not indicating the severity of accidents? Is it the result of pooling categories with low percentages? In view of the numbers in Figure 1, there might be a mistake in line 263.

Thank you very much for your detailed feedback. Following your suggestions, all the errors detected have been corrected.

In conclusion, this paper has the potential for being publishable, but a good re-writing is required.

We are grateful for your comment and have in any case carried out a complete revision of the document.

Reviewer 3 Report

Comments and Suggestions for Authors

This paper analyzes the key factors affecting the occupational accidents related to cranes in Spain in recent decades. The research is very interesting, and the data collection work is full, but the data mining and analysis are not deep enough. In addition, it is suggested to readjust the structure of the paper and highlight research priorities.

1.       The organization of the introduction is chaotic and lacks logic. The classification part of the crane in section 1.2 does not seem to play a role, please explain the significance of the existence of section 1.1.

2.       Section 1.2 wants to express the influencing factors, but does not clearly write the factors affecting the crane accident, so a comprehensive expression is needed. Also, why Section 3 only analyzes a few. Please state the specific reasons for your choice.

3.       The introduction of statistical methods in section 2.3 lacks details, such as significance analysis conditions.

4.       Data sources are introduced in lines 145-146, and the first introduction should clarify the official sources of data.

5.       The conclusion does not clearly state the key factors affecting crane related occupational accidents in the Spanish construction industry.

6.       The supplementary tables for statistical accidents should be supplemented.

Author Response

Ref.:  IJERPH-2652966

Titled: "Key factors in crane-related occupational accidents in the Spanish construction industry (2012-2021)”

Response to Editor and Reviewers

Dear Reviewer   

I would like to thank you for your accurate and detailed revision for improving the paper. In the attached file, you will find a new version of the article, which has been revised in accordance with the recommendations of the editor and reviewers. New comments and explanations according to editor’ suggestions and reviewers’s suggestions have been highlighted in blue (Reviewer #1), yellow (Reviewer #2), or green (Reviewer #3) based on who suggested the changes.  In what follows, we explain how we have responded to and dealt with each of the comments.

Reply to Reviewer #3:

Reviewer #3: This paper analyzes the key factors affecting the occupational accidents related to cranes in Spain in recent decades. The research is very interesting, and the data collection work is full, but the data mining and analysis are not deep enough. In addition, it is suggested to readjust the structure of the paper and highlight research priorities.

Thank you very much for your very constructive comments which will surely help us to improve the article. We will try to respond to all your comments below:

  1. The organization of the introduction is chaotic and lacks logic. The classification part of the crane in section 1.2 does not seem to play a role, please explain the significance of the existence of section 1.1.

The aim was to provide some background information based on the state of the art to make clear the importance of the construction industry in the country's economy and, from there, to detail how accidents at work affect this sector and, more specifically, crane operations in the residential sector.

In section 1.1. it was considered necessary to provide a selection and classification of the different types of cranes according to experts in the field. However, the notification or accident report through the computer system called "Delt@" in Spain does not provide specific information on the type of crane being used in the reported accident, beyond the material agent included in this study.

  1. Section 1.2 wants to express the influencing factors, but does not clearly write the factors affecting the crane accident, so a comprehensive expression is needed. Also, why Section 3 only analyzes a few. Please state the specific reasons for your choice.

Following your suggestions, the title of Section 1.2 is amended to read as follows:

“1.2. Factors influencing crane operation safety in the construction industry”

  1. The introduction of statistical methods in section 2.3 lacks details, such as significance analysis conditions.

Following your suggestions, the following information is expanded with a final paragraph in Section 2.3:

“For this analysis, contingency tables were prepared and the statistical Chi-square value (χ2) was calculated in order to accept or reject the null hypothesis of independence. This statistic associated with a significance level p<0.05 allows us to verify with a confidence level of 95% the relationship of dependence between the variables analysed.”

  1. Data sources are introduced in lines 145-146, and the first introduction should clarify the official sources of data.

Following the reviewer's indications, the following paragraph is included on page 5:

“The data were collected from the Spanish Government's Occupational Accident Statistics corresponding to years 2012-2021”.

  1. The conclusion does not clearly state the key factors affecting crane related occupational accidents in the Spanish construction industry.

We believe that the key factors are the size of the workplace where the crane operator performs his or her duties, the experience of the crane operator on the job, as well as the high level of subcontracting that occurs in the construction industry.

  1. The supplementary tables for statistical accidents should be supplemented

In accordance with the reviewer's indication, in section 3.7 "Main results of the inferential statistical analysis" Table 6 has been modified and completed with statistical information.

Round 2

Reviewer 2 Report

Comments and Suggestions for Authors

The authors have addressed all of my comments; no further comments required.

Comments on the Quality of English Language

Please run  a quick spell check

Reviewer 3 Report

Comments and Suggestions for Authors

 Accept in present form